# A Combined Physical and Mathematical Calibration Method for Low-Cost Cameras in the Air and Underwater Environment

**DOI:** 10.3390/s23042041

**Published:** 2023-02-11

**Authors:** Zhenling Ma, Xu Zhong, Hong Xie, Yongjun Zhou, Yuan Chen, Jiali Wang

**Affiliations:** 1College of Information Technology, Shanghai Ocean University, Shanghai 201306, China; 2School of Electronic and Information Engineering, Shanghai Dianji University, Shanghai 201306, China; 3School of Geodesy and Geomatics, Wuhan University, Wuhan 430079, China; 4School of Naval Architecture, Ocean and Civil Engineering, Shanghai Jiao Tong University, Shanghai 200240, China

**Keywords:** low-cost camera calibration, combined physical distortion models, mathematical approximation, photogrammetry

## Abstract

Low-cost camera calibration is vital in air and underwater photogrammetric applications. However, various lens distortions and the underwater environment influence are difficult to be covered by a universal distortion compensation model, and the residual distortions may still remain after conventional calibration. In this paper, we propose a combined physical and mathematical camera calibration method for low-cost cameras, which can adapt to both in-air and underwater environments. The commonly used physical distortion models are integrated to describe the image distortions. The combination is a high-order polynomial, which can be considered as basis functions to successively approximate the image deformation from the point of view of mathematical approximation. The calibration process is repeated until certain criteria are met and the distortions are reduced to a minimum. At the end, several sets of distortion parameters and stable camera interior orientation (IO) parameters act as the final camera calibration results. The Canon and GoPro in-air calibration experiments show that GoPro owns distortions seven times larger than Canon. Most Canon distortions have been described with the Australis model, while most decentering distortions for GoPro still exist. Using the proposed method, all the Canon and GoPro distortions are decreased to close to 0 after four calibrations. Meanwhile, the stable camera IO parameters are obtained. The GoPro Hero 5 Black underwater calibration indicates that four sets of distortion parameters and stable camera IO parameters are obtained after four calibrations. The camera calibration results show a difference between the underwater environment and air owing to the refractive and asymmetric environment effects. In summary, the proposed method improves the accuracy compared with the conventional method, which could be a flexible way to calibrate low-cost cameras for high accurate in-air and underwater measurement and 3D modeling applications.

## 1. Introduction

Due to their advantages of flexibility and affordability, low-cost cameras have been widely used in close-range photogrammetric applications in air and underwater environments. Typical applications include three-dimensional image reconstruction for important culture/religion sites [1], deformation monitoring of the industrial components [2], topographic mapping of shallow water area [3], and accurate underwater measurements of fish [4], and so on. Camera calibration is essential and vital in the image reconstruction process. Due to small distortion errors, the mathematical relationship between the image space and object space cannot be established rigorously, which results in an accuracy decrease in spatial coordinates. Accordingly, effective distortion compensation models are always expected to achieve the highest-level calibration precision.

Camera calibration using the bundle adjustment method was developed in the early 1970s [5], and was employed in the close-range photogrammetry and reached maturity in the 1980s [6,7,8]. To obtain high-precision calibration results, additional parameters are introduced to compensate for the systematic lens distortion errors [9,10,11]. Some researchers have applied distortion models with significant physical meaning, for example, Brown [5] considered the radial and decentering distortion for the commercial Schneider Symmar lens and it became an accepted standard for the analytical camera calibration; Fraser [12] found that four principal sources of radial distortion, decentering distortion, image plane unflatness, and in-plane image distortion should be considered in the calibration process and the mathematical models were introduced in detail; Remondino and Fraser [13] explained how the chromatic aberration had an impact on the radial distortion model for the consumer-grade digital color cameras, and camera calibration should be concerned yet with the increase of camera types; Luber [14] presented a statistical procedure to choose the most appropriate model automatically and it was suitable for wide-angle fisheye lenses, spherical lenses, and perspective lenses with a calibrating precision of 1 pixel. Some other studies suggested that the camera calibration can be considered as a mathematical approximation problem [15]. For example, Tang et al. [16,17] have verified that the orthogonal Legendre and Fourier polynomials are rigorous and efficient for calibrating the airborne cameras.

Compared to in-air calibration, the influence of light refraction traveling through water, the waterproof housing, and the air must be considered in the underwater camera calibration, rather than calibrating the lens distortions only [18]. Researches show that there are two approaches for underwater camera calibration: one is based on the refraction geometry, in which the light propagation is considered in multimedia and the imaging geometric model introducing refraction is established [19]. Lavest [20] considered the effect of refraction on light ray tracing and proposed a correction to the focal length, and a rescaling of the distortion parameters as a function of the refractive index. Underwater refractive geometry is regarded as a perspective camera looking through parallel flat refractive mediums that can be orientated by the underlying axis related to the number of lays, distances from the camera, and the refractive indices [21,22,23]. The strict refraction imaging model is complicated and there are always some assumptions in the underwater environment [24]. The other method is based on compensating for the refraction effects, in which the standard pinhole-camera model is used to describe the object-image relationship and the systematic refraction errors are additionally compensated in the calibration process [24]. Harvey and Shortis [25,26] performed calibration to compensate the refraction effect on the light path. However, there will always be some systematic errors that are not incorporated into the model.

As mentioned above, both the physical and mathematical approximation models can lead to a successful systematic error compensation and accurate calibration for most cameras used today in photogrammetry [27,28,29]. However, a wide variety of low-cost digital cameras, due to their utility and flexibility, are nowadays employed across air and underwater photogrammetric applications. Accordingly, the influence of light refraction is concerned during the underwater camera calibration process. The various lens distortions and the influence of the underwater environment are difficult to be covered with a universal mathematical model, and the residual distortions may still remain after the calibration using conventional methods. In some precise engineering applications, high accuracies (0.1 pixel or better at the image space) are required [30].

In this paper, we propose a combined physical and mathematical camera calibration method for low-cost cameras, which can adapt to both in-air and underwater environments. Photogrammetric camera calibration can—to a very large extent—be considered as a function approximation problem in mathematics. The distortion can be described by a linear combination of specific basis functions [16,17]. First, the commonly used radial, decentering, and in-plane distortion physical models are applied to describe the image distortions [12] in the proposed method. These models are expressed as high order polynomials, which can be combined with polynomial basis functions. During the in-air or underwater imaging process, the lens distortions, light refraction through multiple mediums, and other environmental factors synthetically influence the image deformation, which can be successively approximated using the polynomials from the point of view of mathematical approximation. Then, these distortion model parameters are estimated with the aid of calibration fixture. Subsequently, the calibration fixture images are geometrically corrected using the estimated distortion parameters. Next, the calibration process is performed again, in which the geometrically corrected calibration fixture images are treated as distorted images, and the distortion models are applied again to describe the distortions. Accordingly, a new set of camera calibration parameters is estimated. Specifically, when the previous camera calibration results (several sets of calibration parameters) are applied to the calibration fixture image geometric correction, in order to avoid multiple resampling degrading image quality, a strategy is proposed to perform resampling just once from the original calibration fixture images. This process is repeated until certain criteria are met. In the end, several sets of distortion parameters are obtained to jointly describe all the camera distortions, along with the obtained camera interior orientation (IO) parameters, acting as the final camera calibration results.

In the following sections, the proposed camera calibration method is introduced in detail, and several experiments of low-cost camera calibration in the air and underwater environment are conducted. The results are analyzed and compared to the conventional camera calibration method (using the Australis model in this case, referring to it as the conventional method).

## 2. The Combined Physical and Mathematical Camera Calibration Method

The key work in this paper is to develop a combined physical and mathematical camera calibration technique that can calibrate all the cameras in the air and underwater environment. All the theory involved in the proposed method will be introduced, including the image distortion description, camera parameter estimation, geometric correction of calibration fixture images, and accuracy assessment (Figure 1).

(1)Image distortion description

The commonly used physical radial, decentering, and in-plane distortion models are integrated to describe the image distortions [12], which can be regarded as the high order polynomials of $\bar{x}$ and $\bar{y}$ in mathematics to successively approximate the image distortions derived from the lens distortions, light refraction through multiple mediums, and other environmental factors, as shown in Equation (Equation 1): (1)$$\left\{\begin{array}{c} \Delta x_{i}={\bar{x}}_{i}\left[k_{1i}\left({{\bar{x}}_{i}}^{2}+{{\bar{y}}_{i}}^{2}\right)+k_{2i}{\left({{\bar{x}}_{i}}^{2}+{{\bar{y}}_{i}}^{2}\right)}^{2}+k_{3i}{\left({{\bar{x}}_{i}}^{2}+{{\bar{y}}_{i}}^{2}\right)}^{3}\right]\\ +p_{1i}\left[\left({{\bar{x}}_{i}}^{2}+{{\bar{y}}_{i}}^{2}\right)+2{{\bar{x}}_{i}}^{2}\right]+2p_{2i}{\bar{x}}_{i}{\bar{y}}_{i}+b_{1i}{\bar{x}}_{i}+b_{2i}{\bar{y}}_{i}\\ \Delta y_{i}={\bar{y}}_{i}[k_{1i}\left({{\bar{x}}_{i}}^{2}+{{\bar{y}}_{i}}^{2}\right)+k_{2i}{\left({{\bar{x}}_{i}}^{2}+{{\bar{y}}_{i}}^{2}\right)}^{2}+k_{3i}{\left({{\bar{x}}_{i}}^{2}+{{\bar{y}}_{i}}^{2}\right)}^{3}]\\ +2p_{1}{\bar{x}}_{i}{\bar{y}}_{i}+p_{2i}\left[\left({{\bar{x}}_{i}}^{2}+{{\bar{y}}_{i}}^{2}\right)+2{{\bar{y}}_{i}}^{2}\right] \end{array}\right.$$
where i=1,2,...,N stands for the number of the camera calibration; $\Delta x_{i}$ and $\Delta y_{i}$ represent the image distortions in the line and sample directions at the *i*th camera calibration; *k*${}_{1i}$, *k*${}_{2i}$, and *k*${}_{3i}$ are the radial distortion parameters at the *i*th camera calibration; *p*${}_{1i}$ and *p*${}_{2i}$ are the decentering distortion parameters at the *i*th camera calibration; *b*${}_{1i}$ and *b*${}_{2i}$ are the in-plane distortion parameters at the *i*th camera calibration; $\bar{x_{i}}=x_{i-1}-x_{0i}$; $\bar{y_{i}}=y_{i-1}-y_{0i}$; *x*${}_{0i}$ and *y*${}_{0i}$ are the principal point offset at the *i*th camera calibration; (*x*_*i*−1_,*y*_*i*−1_) are the image coordinates on the (i−1)th corrected calibration fixture images.

(2)Camera calibration parameters estimation

The imaging geometry of digital cameras in photogrammetry can be modeled according to the principle of a pinhole camera, and the well-known collinearity equation [5] is used to depict the perspective transformation between the image space and object space, as shown in Equation (Equation 2):(2)$$\left\{\begin{array}{c} x_{i-1}-x_{0i}+\Delta x_{i}=-f_{i}\frac{{\bar{Z}}_{1i}}{{\bar{Z}}_{3i}}\\ y_{i-1}-y_{0i}+\Delta y_{i}=-f_{i}\frac{{\bar{Z}}_{2i}}{{\bar{Z}}_{3i}} \end{array}\right.$$
where ($\bar{Z}$${}_{1i}$ $\bar{Z}$${}_{2i}$ $\bar{Z}$${}_{3i}$)${}^{T}$ = R${}_{i}$(*X*-*X*${}_{0i}$ *Y*-*Y*${}_{0i}$ *Z*-*Z*${}_{0i}$)${}^{T}$. (*X*,*Y*,*Z*) are the object coordinates of the Control Points (CPs) in the geographic projection coordinate system; *f*${}_{i}$ is the principal distance at the *i*th camera calibration; R${}_{i}$ stands for the rotation matrix at the *i*th camera calibration, constituted with the attitude angles (φ${}_{i}$, ω${}_{i}$, κ${}_{i}$) of the image; (*X*${}_{0i}$ *Y*${}_{0i}$ *Z*${}_{0i}$) is the perspective center position at the *i*th camera calibration.

Afterwards, the calibration parameters could be solved using the least square estimation method with even CPs, which is widely used in the photogrammetry. The observation equations are obtained when Equation (Equation 2) is linearized using the Taylor series expansion and the second-order and higher-order terms are ignored, as shown in Equation (Equation 3):(3)$$\left\{\begin{array}{c} v_{x_{i-1}}=\frac{dx_{i-1}}{dk_{1i}}\cdot dk_{1i}+\frac{dx_{i-1}}{dk_{2i}}\cdot dk_{2i}+\frac{dx_{i-1}}{dk_{3i}}\cdot dk_{3i}+\frac{dx_{i-1}}{dp_{1i}}\cdot dp_{1i}+\frac{dx_{i-1}}{dp_{2i}}\cdot dp_{2i}+\frac{dx_{i-1}}{db_{1i}}\cdot db_{1i}\\ \;\;\;\;\;\;\;\;\;\;\;+\frac{dx_{i-1}}{db_{2i}}\cdot db_{2i}+\frac{dx_{i-1}}{dX_{0i}}\cdot dX_{0i}+\frac{dx_{i-1}}{dY_{0i}}\cdot dY_{0i}+\frac{dx_{i-1}}{dZ_{0i}}\cdot dZ_{0i}+\frac{dx_{i-1}}{d\varphi_{i}}\cdot d\varphi_{i}+\frac{dx_{i-1}}{d\omega_{i}}\cdot d\omega_{i}\\ \;\;\;\;\;\;\;\;\;\;\;+\frac{dx_{i-1}}{d\kappa_{i}}\cdot d\kappa_{i}+\frac{dx_{i-1}}{dx_{0i}}\cdot dx_{0i}+\frac{dx_{i-1}}{dy_{0i}}\cdot dy_{0i}+\frac{dx_{i-1}}{df_{i}}\cdot df_{i}-\left(x_{i-1}^{0}-x_{i-1}\right)\\ v_{y_{i-1}}=\frac{dy_{i-1}}{dk_{1i}}\cdot dk_{1i}+\frac{dy_{i-1}}{dk_{2i}}\cdot dk_{2i}+\frac{dy_{i-1}}{dk_{3i}}\cdot dk_{3i}+\frac{dy_{i-1}}{dp_{1i}}\cdot dp_{1i}+\frac{dy_{i-1}}{dp_{2i}}\cdot dp_{2i}+\frac{dy_{i-1}}{db_{1i}}\cdot db_{1i}\\ \;\;\;\;\;\;\;\;\;\;+\frac{dy_{i-1}}{db_{2i}}\cdot db_{2i}+\frac{dy_{i-1}}{dX_{0i}}\cdot dX_{0i}+\frac{dy_{i-1}}{dY_{0i}}\cdot dY_{0i}+\frac{dy_{i-1}}{dZ_{0i}}\cdot dZ_{0i}+\frac{dy_{i-1}}{d\varphi_{i}}\cdot d\varphi_{i}+\frac{dy_{i-1}}{d\omega_{i}}\cdot d\omega_{i}\\ \;\;\;\;\;\;\;\;\;\;\;+\frac{dy_{i-1}}{d\kappa_{i}}\cdot d\kappa_{i}+\frac{dy_{i-1}}{dx_{0i}}\cdot dx_{0i}+\frac{dy_{i-1}}{dy_{0}}\cdot dy_{0}+\frac{dy_{i-1}}{df_{i}}\cdot df_{i}-\left(y_{i-1}^{0}-y_{i-1}\right) \end{array}\right.$$
where *v*${}_{x_{i-1}}$ and *v*${}_{y_{i-1}}$ are the observational residuals in the line and sample directions; *dk*${}_{1i}$, *dk*${}_{2i}$, *dk*${}_{3i}$, *dp*${}_{1i}$, *dp*${}_{2i}$, *db*${}_{1i}$ and *db*${}_{2i}$ are the corrections to the distortion parameters; *dx*${}_{0i}$, *dy*${}_{0i}$ and *df_i_* are the corrections to the camera IO parameters; *dX*${}_{0i}$, *dY*${}_{0i}$, *dZ*${}_{0i}$, *d*$\varphi_{i}$, *d*$\omega_{i}$, *d*$\kappa_{i}$ are the corrections to the exterior orientation elements; *x*${}_{i-1}$${}^{0}$ and *y*${}_{i-1}$${}^{0}$ are the nominal line and sample coordinates of CPs computed by Equation (Equation 2).

A matrix version of the observation Equation (Equation 3) is shown in Equation (Equation 4):(4)$$V_{i-1}=A_{i}t_{i}-L_{i-1};\;\;\;\;\;\;\;\;\;\;\;\;\;\;\;\;\;\;\;\;\;\;\;\;\;\;\;\;\;\;\;\;\;\;\;\;\;\;\;\;P=I$$
where *V*${}_{i-1}$ refers to the observational residual vector; **A**${}_{i}$ contains the partial derivatives of the distortion parameters, camera IO parameters and the exterior orientation elements; *t*${}_{i}$ is the vector of corrections to the calibration parameters, including distortion parameters, camera IO parameters, and the exterior orientation elements; *L*${}_{i-1}$ is the observation vector of points’ image coordinates; **P** stands for the weights of CPs on the images, and they are observed artificially or by the image matching method at the same accuracy, therefore the weight matrix is set to unit matrix **I**.

The corrections of the calibration parameters will be obtained by least squares, as shown in Equation (Equation 5):(5)$$t_{i}={\left(A_{i}^{T}PA_{i}\right)}^{-1}A_{i}^{T}PL_{i-1}$$

In the linearization procedure, the coefficients of distortion parameters’ correction retain the first-order terms only, and the initial values of these unknowns are set to 0, therefore an iterative process is necessary to determine the distortion parameters. The solution of distortion parameters is shown in Equation (Equation 6):(6)$$T_{i}=T_{i-1}+q_{i}$$
where *T*${}_{i}$ = [*k*${}_{1i}$, *k*${}_{2i}$, *k*${}_{3i}$, *p*${}_{1i}$, *p*${}_{2i}$, *b*${}_{1i}$, *b*${}_{2i}$] represents the obtained distortion parameters; *T*${}_{i-1}$ is the previous distortion parameters at the (i−1)th camera calibration; *q*${}_{i}$ is the obtained corrections of the distortion parameters at the *i*th calibration process.

In order to avoid the correlations among the calibration parameters, the calibration fixtures are captured at various scales and different stations, forming a robust network containing short baseline image sequences that can provide benefits to increase the redundant observations and produce strong intersections [30].

(3)Geometric correction of the calibration fixture images

The calibration fixture images are geometrically corrected using the previous camera calibration results. In order to avoid multiple resampling degrading image quality, a strategy is proposed to perform resampling just once from the original calibration fixture images (Figure 2). The pixels on the corrected calibration fixture images (Figure 2c) are directly reversed to acquire the corresponding pixels on the original calibration fixture images (Figure 2a) using the previous *m* sets of distortions, as shown in Figure 2 and Equation (Equation 7). Then, the resampling is performed on the original calibration fixture images using the cubic convolution method to obtain the grey value of these pixels.
(7)$$\left\{\begin{array}{c} x^{\prime }=x-x_{0}+\Delta x_{1}+\Delta x_{2}...+\Delta x_{m}\\ y^{\prime }=y-y_{0}+\Delta y_{1}+\Delta y_{2}...+\Delta y_{m} \end{array}\right.\;\;\;\left(m\geq 1\right)$$
where (x’, y’) are the corrected image coordinates; *x*${}_{0}$ and *y*${}_{0}$ are the principal point offset; (*x*, *y*) are the image coordinates on the original calibration fixture images. Subsequently, the corrected calibration fixture images are taken as the distorted images to repeat the calibration process, and so on, until certain criteria are met.

At the end, several sets of distortion parameters (Δx${}_{1}$, Δx${}_{2}$... Δx${}_{n}$ and Δy${}_{1}$, Δy${}_{2}$... Δy${}_{n}$) are used to jointly describe all the camera distortions, which, along with the obtained camera IO parameters, act as the final camera calibration results.

(4)Accuracy assessment

CPs that acted as check points are used for external accuracy assessment in the object space. The nominal object coordinates of these check points are calculated using Equation (Equation 2) with the direct intersection method. The Root Mean Square Error (RMSE) in the object space is computed based on the discrepancies between the truth values and nominal values of the check points, shown in Equation (Equation 8):(8)$$\;\left\{\begin{array}{c} \mu_{X}=\sqrt{\frac{\sum {\left(X_{t}-X_{c}\right)}^{2}}{r}}\\ \mu_{Y}=\sqrt{\frac{\sum {\left(Y_{t}-Y_{c}\right)}^{2}}{r}}\\ \mu_{Z}=\sqrt{\frac{\sum {\left(Z_{t}-Z_{c}\right)}^{2}}{r}}\\ \mu_{P}=\sqrt{{\mu_{X}}^{2}+{\mu_{Y}}^{2}+{\mu_{Z}}^{2}} \end{array}\right.$$
where $\mu_{X}$, $\mu_{Y}$ and $\mu_{Z}$ represent the RMSE of check points for the corresponding three directions; *r* refers to the number of check points; *X*${}_{t}$, *Y*${}_{t}$, *Z*${}_{t}$ are the truth coordinates of the check points; *X*${}_{c}$, *Y*${}_{c}$, *Z*${}_{c}$ are the nominal coordinates of the check points. $\mu_{P}$ is used to state the accuracy in the object space in the following discussions.

## 3. Experimental Results and Discussion

### 3.1. Calibration Fixture

In this study, the calibration fixtures of a 3D test field and a 2D planar calibration board are used to assist the camera calibration. The 3D test field is established on the shape invariant brick wall of a building and fits the in-air camera calibration. The size of each brick is 24 cm × 6 cm. A total of 188 evenly distributed corner points of the bricks are taken as the CPs and check points. The object coordinates of these points are measured by the total station and the measurement accuracies are 2 mm. The established 3D test field of the brick wall and the distributed CPs are shown in Figure 3. The black dots represent the CPs. The 2D planar calibration board is applied for camera calibration in the underwater environment. The diffused reflective planar calibration board is used, which is sized at 400 mm × 400 mm and consists of 625 circular targets (dots). Dots are chosen as targets owing to the isotropy of circles, and the CPs are located in the center. The dots are evenly positioned across 25 rows and 25 columns. Each individual circular dot has a diameter of 7.5 mm, and two neighboring dots are 15 mm apart. The measurement accuracy is 0.01 mm. The 2D diffused reflective planar calibration board and circular targets are shown in Figure 4.

### 3.2. Tests of Low-Cost Camera Calibration

In this study, two types of low-cost cameras are calibrated in the air and underwater environment using the proposed camera calibration method, including a full-frame digital single-lens reflex camera (Canon EOS 5Dsr) with a prime lens of TS-E 24 mm 1:3.5 L II and an action camera (GoPro Hero 5 Black) operated in photo mode with a narrow field of view. The Canon camera deploys a Complementary Metal Oxide Semiconductor (CMOS) imaging sensor and the size is 36 mm × 24 mm with a resolution of 8688 pixels × 5792 pixels. The given focal length of the lens is 24 mm. The GoPro carries a 1/2.3-inch image sensor with a resolution of 12 megapixels, and is equipped with a wide-angle lens, leading to large distortions for the captured images. The in-air calibration is performed using both cameras, and the underwater calibration is carried out with the GoPro camera.

#### 3.2.1. Calibration Tests in the Air

The 3D test field was employed to calibrate the Canon and GoPro cameras in the air. Forty-five images were acquired via the Canon camera, which were captured at two scales, 1.80 m and 2.60 m away from the test field. Seventeen images were acquired by the GoPro camera at three distances of 1.10 m, 1.65 m, and 2.50 m. In order to avoid introducing other systematic errors, each image was filled with the calibration fixture, and no other object entered the picture.

A total of 98 CPs and 61 CPs evenly distributed in the test field are used for the Canon and GoPro calibration parameters estimation, respectively. A total of 90 CPs and 127 CPs are used for the Canon and GoPro calibration accuracy assessment, respectively. On the original images, the CPs are plotted manually. For the following calibrations, the CPs on the corrected images are measured automatically according to Equation (Equation 7) without extra work. The radial distortion, decentering distortion, and in-plane distortion of the two cameras at each calibration are profiled separately, shown in Figure 5, Figure 6, Figure 7, Figure 8, Figure 9 and Figure 10. The calibrated IO parameters of the two cameras and the calibration accuracy are shown in Table 1. The calibration results of the conventional method (using the Australis model once) are exactly the same as that of the proposed method at the first calibration.

From the Canon in-air calibration results (Figure 5, Figure 6 and Figure 7, Table 1), the maximum radial, decentering, and in-plane distortion of the camera are 104 μm, 14 μm and 20 μm, respectively (Figure 5, Figure 6 and Figure 7). These maximum values decrease to 1.14 μm, 1.15 μm and 0.4 μm after the second calibration, which indicates that the residual distortions remained after the calibration with the conventional method (the first calibration). The accuracy shows that compared with the conventional method, the proposed method achieves a slight accuracy improvement of 2.8 μm (0.24%) (Table 1) after the residual distortions are calibrated.

From the GoPro in-air calibration results (Figure 8, Figure 9 and Figure 10, Table 1), the maximum radial, decentering, and in-plane distortion of the camera are 0.72 mm, 0.01 mm, and 0.07 mm, respectively (Figure 8, Figure 9 and Figure 10), which are calibrated by the conventional method (the first calibration). These maximum values decrease to 0.77 μm, 9.33 μm, and 0.4 μm after the second calibration, which indicates that the residual distortions, especially most decentering distortions, remained after the calibration with the conventional method (the first calibration). The proposed method achieves an accuracy improvement of 0.2 mm (3.88%) compared with the conventional method (Table 1) after the residual distortions are calibrated.

In summary, the Canon and GoPro in-air calibration results show that the distortions gradually grow larger with the increase of the radial distance from the center of the image. The in-plane distortion is distributed unequally on the image. All the residual distortions of the Canon and GoPro cameras are decreased as the number of calibration increases and approach 0 after four calibrations, which illustrates that all the distortions have been fully calibrated by the proposed method. Meanwhile, the stable camera IO parameters are obtained after the repeated calibration processes. From the magnitudes of distortions, the lens distortion is dominated by the radial distortion for the two cameras [18], and the GoPro lens has distortions seven times larger than the Canon lens. Most distortions of the Canon lens have been detected with the conventional method (the first calibration), while a large number of decentering distortions for the GoPro lens still exist (Figure 9). This indicates that the Australis model cannot cover the distortions of all the cameras, with which the Canon lens is better described than the GoPro lens.

#### 3.2.2. Calibration Tests in the Underwater Environment

Due to the smaller field of view and close range caused by the attenuation of light in the water, the 2D planar calibration board was employed to calibrate the GoPro camera in the underwater environment. Because of refraction, underwater images exhibit non-linear distortions related to the distance from the object to the camera [31]. Therefore, two underwater camera calibration tests were executed with two distances, respectively. The first test captured 33 images at a distance of 0.14 m away from the calibration board. A total of 33 CPs and 45 CPs are used to calculate the calibration parameters and assess the calibration accuracy, respectively. The second test acquired 10 images at a distance of 0.50 m away from the calibration board. A total of 24 CPs and 30 CPs are used to calculate the calibration parameters and assess the calibration accuracy, respectively. The calibration results are shown in Table 1 and the distortion profiles are shown in Figure 11, Figure 12, Figure 13, Figure 14, Figure 15 and Figure 16.

(1)Underwater camera calibration at a distance of 0.14 m away from the calibration board

From the GoPro underwater calibration results (Figure 11, Figure 12 and Figure 13, Table 1), the calibrated maximum radial, decentering, and in-plane distortion are 0.11 mm, 0.002 mm, and 25 μm, respectively. These maximum values decrease to 5 μm, 1 μm and 0.06 μm after the second calibration, which are the residual distortions that remained after the calibration with the conventional method. All the residual distortions can be detected by the proposed method. The accuracy shows a rise from 3.8451 mm to 3.8215 mm, indicating that the proposed method improves the accuracy by 0.02 mm (0.61%) compared with the conventional method (Table 1). It is similar to the in-air calibration results, in which the stable camera IO parameters are obtained after the repeated calibration processes.

(2)Underwater camera calibration at a distance of 0.50 m away from the calibration board

From the second GoPro underwater calibration test (Figure 14, Figure 15 and Figure 16, Table 1), the calibrated maximum radial, decentering, and in-plane distortion are 0.16 mm, 0.16 mm, and 0.5 μm, respectively. These maximum values decrease to 17 μm, 1.6 μm, and 0.3 μm after the second calibration. All the residual decentering distortions and in-plane distortions are detected after the second calibration. The residual radial distortions are decreased to 0 after the third calibration. The accuracy shows a rise from 3.4860 mm to 3.4650 mm, indicating that the proposed method improves the accuracy of 0.021 mm (0.60%) compared with the conventional method (Table 1). Similar to the in-air and first underwater calibration results, the stable camera IO parameters are obtained after the repeated calibration processes.

In summary, the GoPro underwater calibration results show variations by distance from the camera to the calibration board. The distortions at a distance of 0.14 m are dominated by the radial distortions, and all the distortions can be calibrated by four repeated calibrations. The radial and decentering distortions are major distortions at a distance of 0.50 m, and all the distortions can be detected by three repeated calibrations. The stable camera IO parameters are obtained after the calibration processes.

Compared to the in-air calibration, the final GoPro camera calibration results in the underwater environment (Table 1) indicate that the achieved principal distance in the underwater environment is different from that obtained in the air, because the light rays refract when traveling from the water to the air, which leads to a change of the principal distance. The shifts of the principal point and the distortions in the underwater environment appear smaller than that in the air, which is caused by the refractive and asymmetric environment effects absorbed by the radial, decentering, and in-plane distortions [18].

In summary, Canon EOS 5Dsr and GoPro Hero 5 Black were calibrated for the the air and underwater environment. The calibration results illustrate that the residual distortions remained with the conventional method. When using the proposed method, the distortions decrease gradually throughout the repeated calibration processes and can be negligible at last. Meanwhile, stable elements of camera IO can be achieved after several calibrations. Desirable results were obtained due to the repetitive use of the physical distortion models during the calibration process. Photogrammetric calibration is regarded as a function approximation problem from the perspective of mathematics [16], and any univariate function can be approximated by a polynomial of sufficiently high degree according to the Weierstrass Theorem [17]. The distortion model combined with radial, decentering, and in-plane physical distortion models in this paper is actually a polynomial with multiple degrees, which is applied to successively approximate the distortions constantly during the calibration processes. Compared to the in-air calibration, the underwater calibration is affected by the complicated underwater environment, and it is difficult to describe these factors with various mathematical models owing to the coupling effects and correlations. The results indicate that the proposed camera calibration method is applicable for the underwater camera calibration as well.

### 3.3. Underwater Application Example

The calibrated camera can be used in three-dimensional reconstruction and measurement of the objects using the photogrammetric method. In this paper, the calibrated GoPro at a distance of 0.5 m in the underwater environment (Figure 14, Figure 15 and Figure 16, Table 1) was used to carry out the underwater object three-dimensional reconstruction and measurement.

Four images covering the bottom of the pool are acquired with GoPro at a depth of 0.50 m, which have about (80%) overlap, as shown in Figure 17. The offset angle between the camera movement direction and the horizontal direction is less than 3${}^{\circ }$.

These four images are geometrically corrected using three sets of calibrated distortion parameters (Figure 14, Figure 15 and Figure 16), and the corrected images are shown in Figure 18. The proposed strategy is used to perform the image resampling (Figure 2). The pixels on the corrected underwater images are directly reversed to acquire the corresponding pixels on the original images using three sets of distortions. Then, the resampling is performed on the original images using the cubic convolution method to obtain the grey value of these pixels.

Three stereo pairs are formed by the four corrected images. The first pair is made up of the corrected image Figure 18a and the corrected image Figure 18b. The second pair is composed of the corrected image Figure 18b and the corrected image Figure 18c. The corrected image Figure 18c and the corrected image Figure 18d form the third pair. About 100 tie points are matched on each image pair using the manual and semi-automatic matching method. The relative orientation is carried out with the image pairs to make the corresponding rays intersect, in which the corrected image Figure 18b is taken as the reference image. Meanwhile, the relationship between the corrected images are established by the calculated exterior orientation parameters. Then, the disparity maps are produced using the oriented image pairs through the dense matching technique (Figure 19).

Three-dimensional point clouds of the scene are obtained using the disparity maps through the forward intersection method. Afterwards, the digital surface model (DSM) of the scene can be acquired by point clouds interpolation (Figure 20a). The corrected images are then rectified with DSM to generate orthophoto (Figure 20b). The three-dimensional model can be reconstructed by DSM and orthophoto. There is no absolute reference at the bottom of the pool, therefore only relative elevation and horizontal information can be obtained from DSM and orthophoto.

An underwater brick is used to carry out the photogrammetric measurement. The true size of the experimental brick is 200.00 mm (length) × 100.00 mm (width) × 50.00 mm (height). Ten points on the brick are chosen to perform the three-dimensional information measurement. Five lines are constituted by these points. They are line 01–02, line 03–04, line 05–06, line 07–08, and line 09–10, respectively. Line 07–08 and line 09–10 stand for the brick length. Line 01–02, line 03–04, and line 05–06 stand for the brick width. The point and line distribution is shown in Figure 21. The three-dimensional coordinates of these points are calculated by forward intersection using the disparity maps and exterior orientation parameters. The length of line is the distance between two points, which can be determined by three-dimensional coordinates of the points. There is no absolute reference at the bottom of the pool, therefore only relative geometric information can be obtained. The brick length and width are calculated by scale conversion using the relative calculated results. The measurement accuracy is assessed by the true value and the calculated value of the brick length and width.

The GoPro has been calculated at a distance of 0.5 m in the underwater environment using the conventional method and the proposed camera calibration method (Figure 14, Figure 15 and Figure 16, Table 1). The calibrated results were applied in the brick size measurement. The measurement accuracy is shown in Table 2.

The measurement accuracy acquired by the proposed method is 1.24 mm, which is superior to the accuracy obtained by the conventional method (Table 2). The proposed camera calibration method can detect almost all the image distortions, and the systematic distortion error can be compensated for in the geometric correction process to improve the subsequent photogrammetric steps.

## 4. Conclusions

In this paper, a combined physical and mathematical calibration method for low-cost cameras in the air and underwater environment is proposed. In the proposed method, the physical radial, decentering, and in-plane distortion models are integrated to describe the camera distortions. The combination is a high-order polynomial, which can be considered as basis functions to successively approximate the distortions during the calibration processes from the point view of mathematical approximation. Then, the proposed method uses the least squares estimation method with even CPs to estimate the camera calibration parameters. Next, the original calibration fixture images are geometrically corrected using the estimated distortion parameters. Unlike the conventional methods, the proposed method performs the above calibration in a repeated fashion: treating the geometrically corrected images as input distorted images, re-applying the distortion model, estimating a new set of camera parameters, and geometrically correcting the already corrected images, and so on, until certain criteria are met. Specifically, when the previous camera calibration results (*m* sets of calibration parameters) are applied to the image geometric correction, in order to avoid multiple resampling degrading image quality, a strategy is proposed to perform resampling just once from the original calibration fixture images. At the end, several sets of distortion parameters and stable camera IO parameters are obtained, which act as the final camera calibration results and are used to describe all the image distortions.

The in-air calibration results with Canon EOS 5Dsr and GoPro Hero 5 Black demonstrate that the distortions of these cameras vary greatly in magnitude. The Australis model cannot cover all the distortions, and the residual distortions still remained with the conventional method. While using the proposed method, all the final distortion parameters are close to 0 and can be negligible after several calibrations, meanwhile, stable elements of IO can be achieved at last. The underwater calibration with GoPro Hero 5 Black indicates that the calibration parameters in the underwater environment appear different from that in the air, owing to the refractive and asymmetric environment effects. The proposed method is applicable for the underwater camera calibration as well. In summary, the calibration experimental results demonstrate that the proposed method is flexible to use and has potentials in practice in order to achieve high three-dimensional measurement accuracies for both in-air or underwater applications.

Further study would be focused on applying the proposed method in the situation where no control point is available, and investigating how the proposed method can be used in triangulation while camera calibration and block adjustment are performed jointly. Recently, our research team is establishing the refraction physical imaging model in the underwater environment, in which the object distance is taken as one of the parameters.

## Figures and Tables

**Figure 1 sensors-23-02041-f001:**
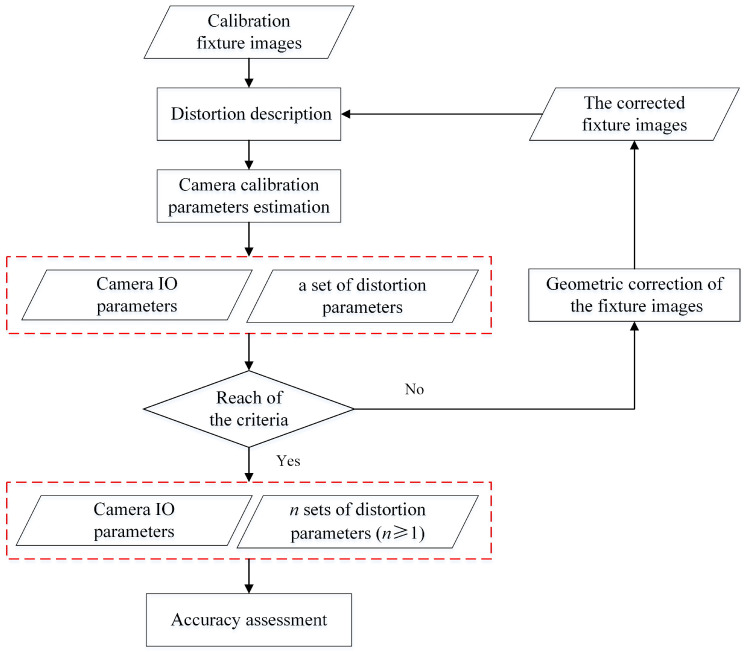
The workflow of the proposed calibration method for a low-cost camera.

**Figure 2 sensors-23-02041-f002:**
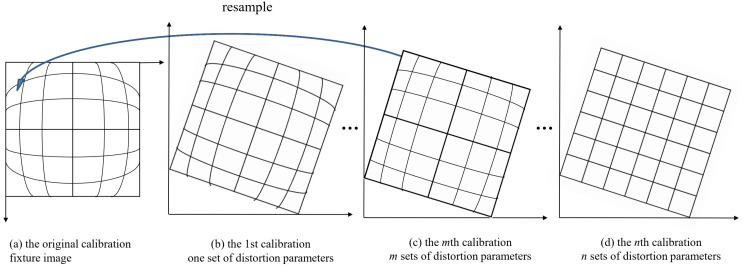
The resampling method from the original calibration fixture image.

**Figure 3 sensors-23-02041-f003:**
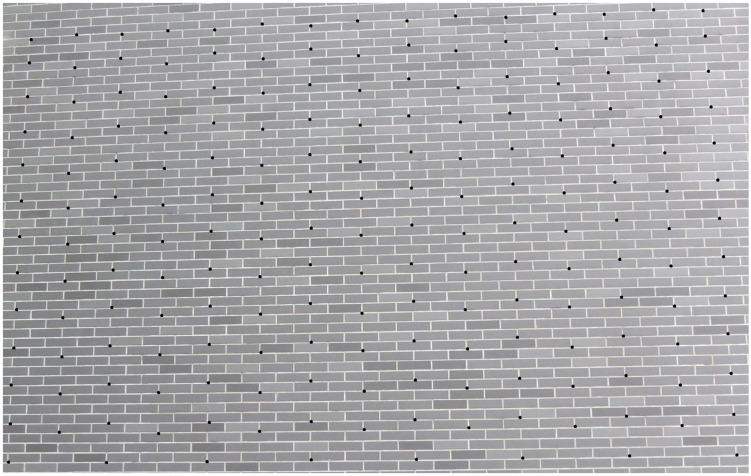
The established 3D test field of the brick wall and the evenly distributed CPs (black dots).

**Figure 4 sensors-23-02041-f004:**
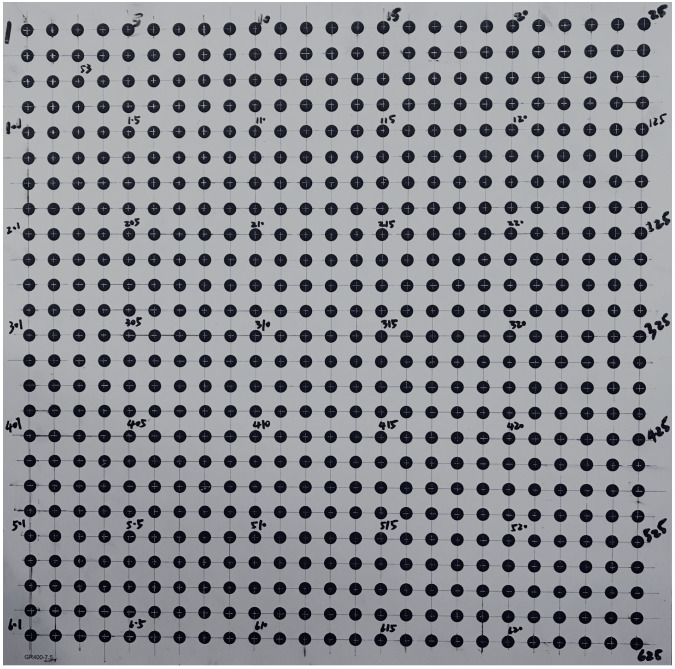
The 2D diffused reflective planar calibration board and circular targets.

**Figure 5 sensors-23-02041-f005:**
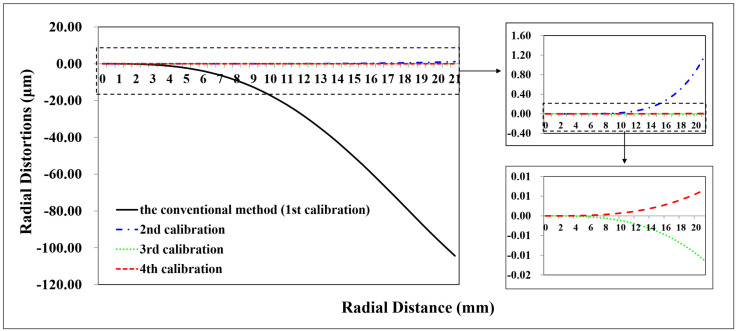
Radial distortion profiles at different radial distances for Canon in the air calibrated using the conventional method and the proposed method at different calibrations.

**Figure 6 sensors-23-02041-f006:**
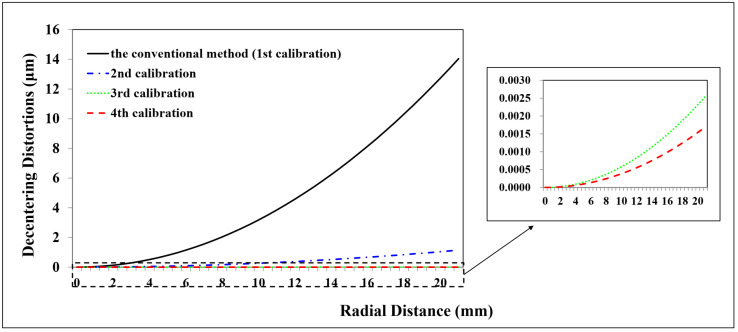
Decentering distortion profiles at different radial distances for Canon in the air calibrated using the conventional method and the proposed method at different calibrations.

**Figure 7 sensors-23-02041-f007:**
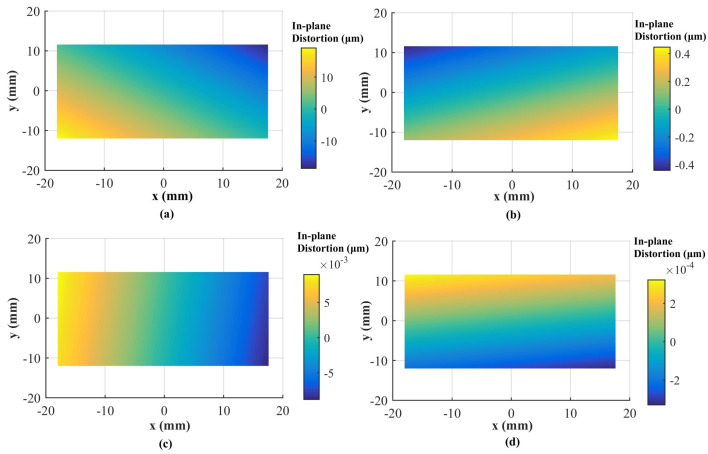
In-plane distortions for Canon in the air calibrated using the conventional method and the proposed method at different calibrations: (**a**) the conventional method (the first calibration); (**b**) the second calibration; (**c**) the third calibration; (**d**) the fourth calibration.

**Figure 8 sensors-23-02041-f008:**
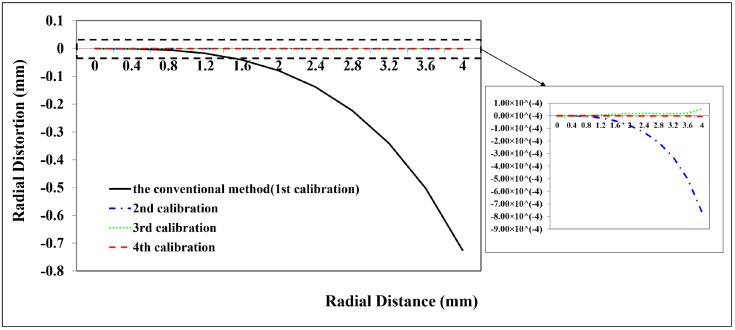
Radial distortion profiles at different radial distances for GoPro in the air calibrated using the conventional method and the proposed method at different calibrations.

**Figure 9 sensors-23-02041-f009:**
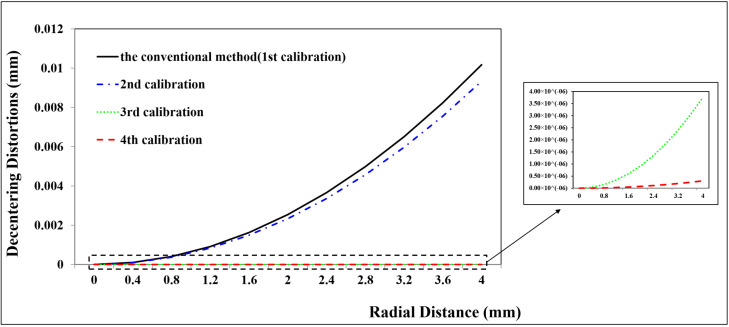
Decentering distortion profiles at different radial distances for GoPro in the air calibrated using the conventional method and the proposed method at different calibrations.

**Figure 10 sensors-23-02041-f010:**
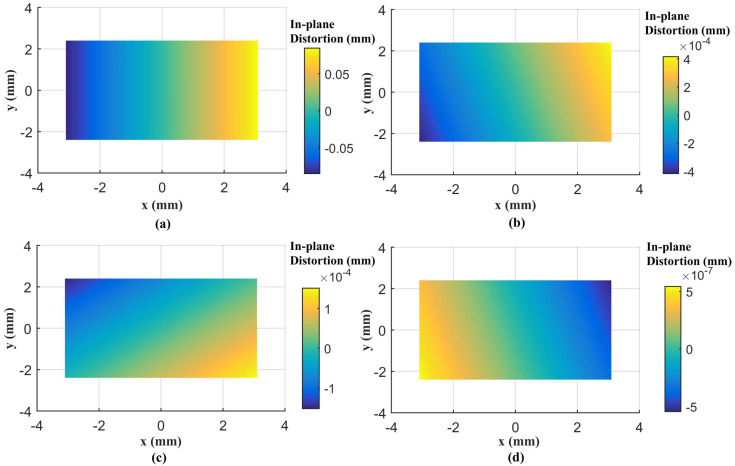
In-plane distortions at different calibrations for GoPro in the air: (**a**) the first calibration; (**b**) the second calibration; (**c**) the third calibration; (**d**) the fourth calibration.

**Figure 11 sensors-23-02041-f011:**
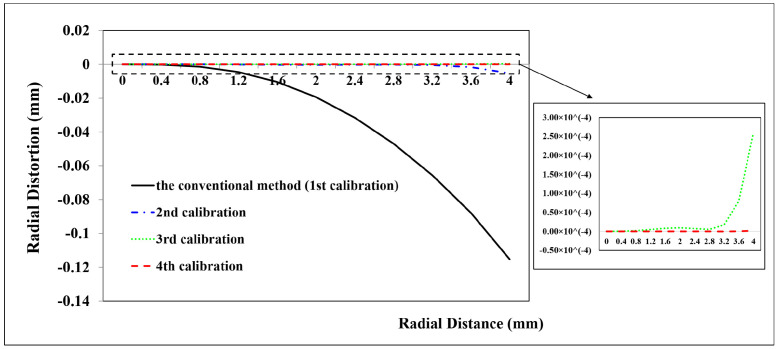
Radial distortion profiles at different radial distances for GoPro in the underwater environment (0.14 m away from the calibration board) calibrated using the conventional method and the proposed method at different calibrations.

**Figure 12 sensors-23-02041-f012:**
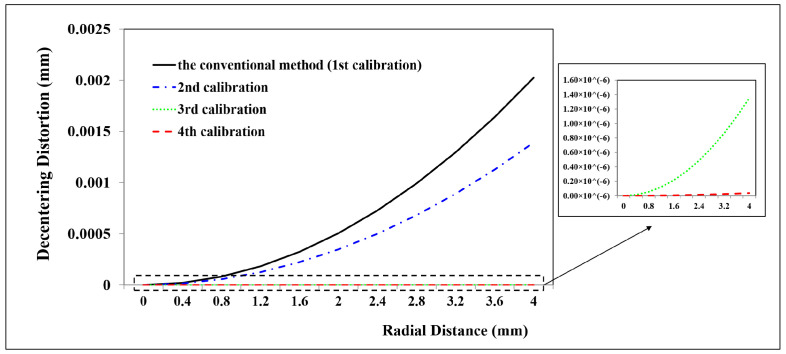
Decentering distortion profiles at different radial distances for GoPro in the underwater environment (0.14 m away from the calibration board) calibrated using the conventional method and the proposed method at different calibrations.

**Figure 13 sensors-23-02041-f013:**
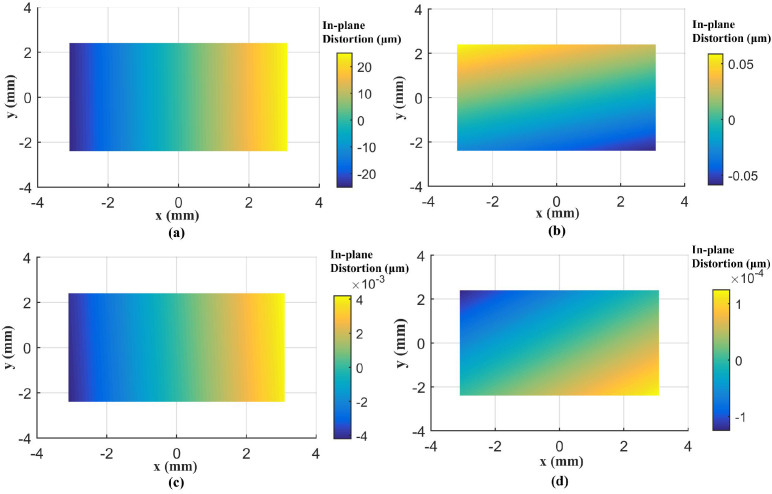
In-plane distortions at different calibrations for GoPro in the underwater environment (0.14 m away from the calibration board): (**a**) the first calibration; (**b**) the second calibration; (**c**) the third calibration; (**d**) the fourth calibration.

**Figure 14 sensors-23-02041-f014:**
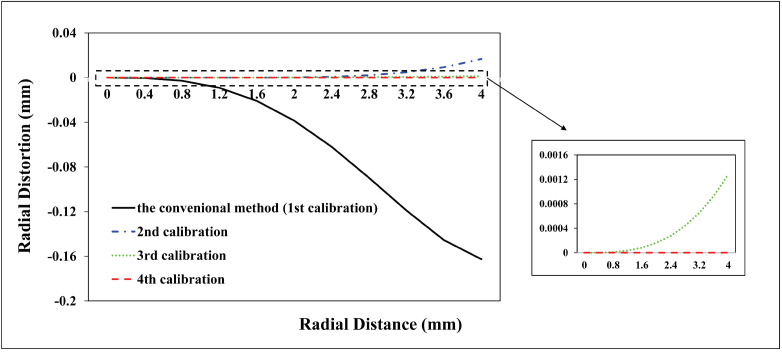
Radial distortion profiles at different radial distances for GoPro in the underwater environment (0.5 m away from the calibration board) calibrated using the conventional method and the proposed method at different calibrations.

**Figure 15 sensors-23-02041-f015:**
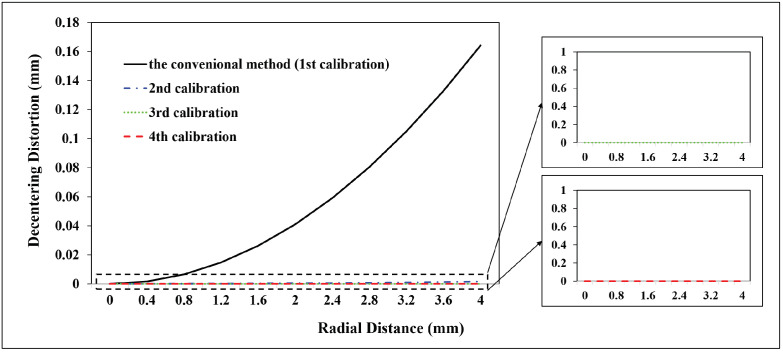
Decentering distortion profiles at different radial distances for GoPro in the underwater environment (0.5 m away from the calibration board) calibrated using the conventional method and the proposed method at different calibrations.

**Figure 16 sensors-23-02041-f016:**
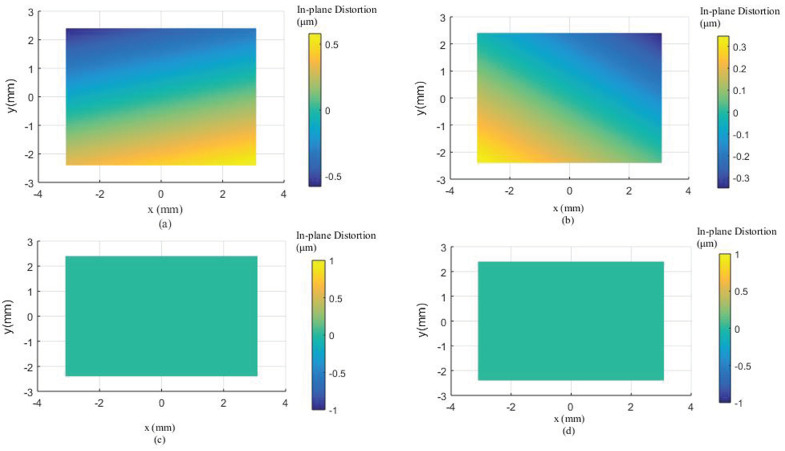
In−plane distortions at different calibrations for GoPro in the underwater environment (0.5 m away from the calibration board): (**a**) the first calibration; (**b**) the second calibration; (**c**) the third calibration; (**d**) the fourth calibration.

**Figure 17 sensors-23-02041-f017:**
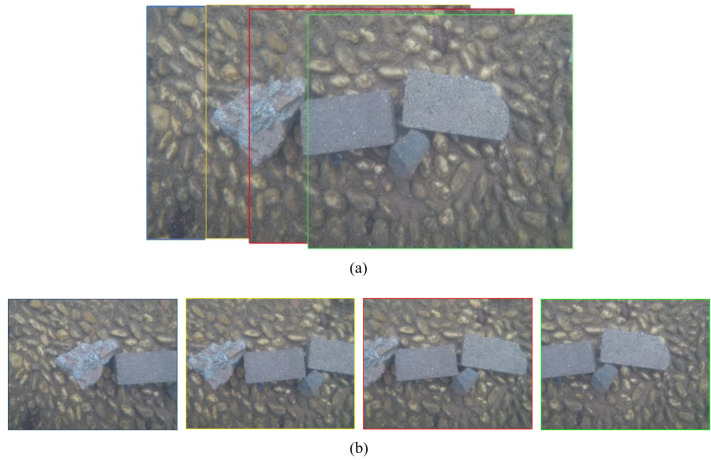
The captured four underwater images: (**a**) the scene covered by four overlapping images; (**b**) four original underwater images.

**Figure 18 sensors-23-02041-f018:**
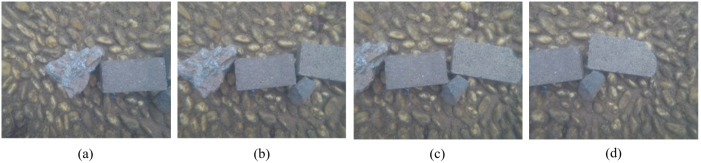
The four corrected underwater images: (**a**) the first image; (**b**) the second image; (**c**) the third image; (**d**) the fourth image.

**Figure 19 sensors-23-02041-f019:**
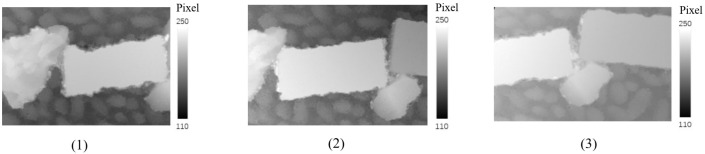
The generated disparity maps: (**1**) the first disparity map; (**2**) the second disparity map; (**3**) the third disparity map.

**Figure 20 sensors-23-02041-f020:**
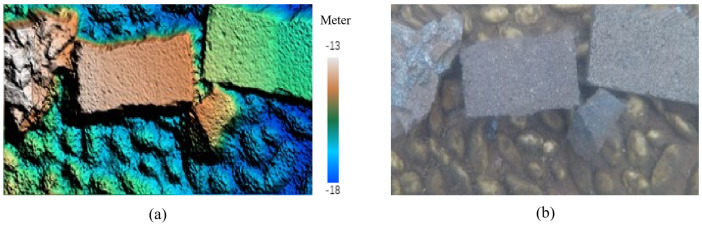
Digital surface model (**a**) and orthophoto (**b**).

**Figure 21 sensors-23-02041-f021:**
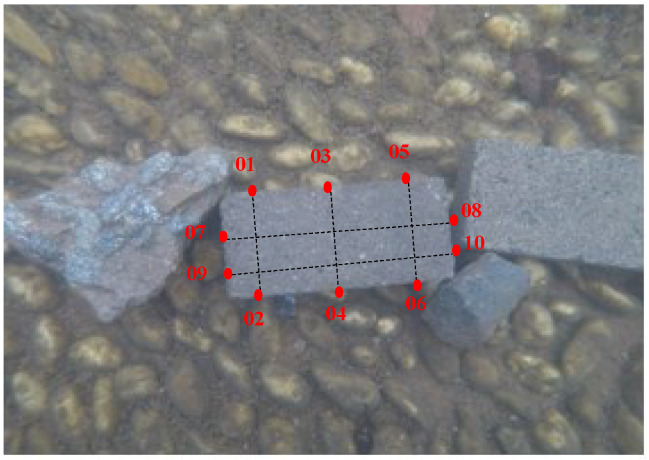
Measurement points and lines distribution.

**Table 1 sensors-23-02041-t001:** The calibrated IO parameters of the two cameras with the conventional method and the proposed method in the air and underwater environment.

Calibration Result		Conventional	Proposed Method (Number of Iterations)			
		Method	1	2	3	4
**Canon in the air**	*f* (mm)	24.2755	24.2755	24.2772	24.2771	24.2771
	*x*${}_{0}$ (mm)	−0.1238	−0.1238	−0.1237	−0.1237	−0.1237
	*y*${}_{0}$ (mm)	0.1527	0.1527	0.1535	0.1534	0.1534
	RMSE (mm)	1.1384	1.1384	1.1384	1.1356	1.1356
**GoPro in the air**	*f* (mm)	5.4432	5.4432	5.4526	5.4576	5.4576
	*x*${}_{0}$ (mm)	0.0472	0.0472	0.0501	0.0505	0.0505
	*y*${}_{0}$ (mm)	0.0452	0.0452	0.0524	0.0541	0.0541
	RMSE (mm)	5.7559	5.7559	5.5365	5.5319	5.5324
**GoPro in the underwater environment (0.14 m)**	*f* (mm)	7.2802	7.2802	7.2650	7.2652	7.2652
	*x*${}_{0}$ (mm)	0.0332	0.0332	0.0317	0.0317	0.0317
	*y*${}_{0}$ (mm)	0.0405	0.0405	0.0414	0.0413	0.0413
	RMSE (mm)	3.8451	3.8451	3.8215	3.8215	3.8215
**GoPro in the underwater environment (0.50 m)**	*f* (mm)	3.9281	3.9281	3.9284	3.9277	3.9277
	*x*${}_{0}$ (mm)	0.0378	0.0378	0.0375	0.0374	0.0374
	*y*${}_{0}$ (mm)	0.0431	0.0431	0.0398	0.0398	0.0398
	RMSE (mm)	3.4860	3.4860	3.4650	3.4650	3.4650

**Table 2 sensors-23-02041-t002:** Measurement accuracy comparison between the conventional camera calibration method and the proposed camera calibration method.

	Conventional Method	Proposed Method
Measurement accuracy (mm)	5.00	1.24

## Data Availability

Data are available upon request.

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
