# Peer review of "A Combined Physical and Mathematical Calibration Method for Low-Cost Cameras in the Air and Underwater Environment"

_sensors, 2023, doi:10.3390/s23042041_

Round 1

Reviewer 1 Report (Previous Reviewer 2)

In my first review I asked for evaluation of the calibration improvement by independent measurement data. I cannot see that you included such data into the manuscript. Additionally, an application example would confirm your statement concerning calibration improvement and derived conclusion.

In your response letter you told me that you are familiar with the properties of distortion in the underwater case and that you know that refraction distortions are different at the varying distances. However, this should also be reflected in your manuscript including the cited references. Accordingly, you should provide at least examples of undistorted images or data from underwater recordings at several significantly different distances.

Hence, for publication these lacks must be necessarily closed.

Author Response

Thanks to you for your good comments. Your suggestions have helped a lot to improve our paper. We have revised the manuscript according to your suggestions. The revised part is highlighted in blue.

Calibration fixture with high-precision Control Points is used for camera calibration, in which sufficient Control Points are used to calculate the calibration parameters and the rest is used for accuracy assessment. We have stated clearly the number of Control Points used for parameter calculation and accuracy assessment in the four calibration tests. The statements are shown in Line 217-219 and Line 262-265, marked in blue. Additionally, the distortions are reduced to minimum using our proposed method, which can also validate the camera calibration superiority.

We have cited the reference [31] about the relationship between refraction and distance. We have provided an underwater calibration test with the depth of 0.5 m in section 3.2.2. The calibrated GoPro has been used to capture the scene of pool bottom. Three dimensional reconstruction and measurement are carried out and taken as an application example. This part is new added in the revised paper, as shown in section 3.3.

We tried our best to improve the manuscript and made some changes in the manuscript. These changes will not influence the content and framework of the paper. And here we did not list the changes but marked in blue in revised paper. We appreciate for your warm work earnestly, and hope that the correction will meet with approval. Once again, thank you very much for your comments and suggestions.

Reviewer 2 Report (New Reviewer)

It is my first review of this manuscript in which the authors proposed a calibration method that combines both physical and mathematical models along with iterative correction for low-cost cameras. Test results show the proposed method is effective and performs better than traditional method. The manuscript is acceptable after correcting some grammar problems. I have annotated all the corrections in the attached PDF file.

Author Response

We are very sorry for our incorrect writing, and we have made corrections according to your annotation. The correction is highlighted in blue. We appreciate for your warm work earnestly, and hope that the correction will meet with approval.

Reviewer 3 Report (New Reviewer)

The author presents "A Combined Physical and Mathematical Camera Calibration Method for Low-cost Cameras in the Air and Underwater Environment" is well structured and written, i suggested  accept manuscript after few changes in references by changing reference 9,11,14,21 and 23 and provide the quality reference rather than conference proceedings.

Author Response

As reviewer suggested that it is indeed better to give some quality reference. Reference 9, 21 and 23 have been replaced by journal papers. Reference 11 is a journal paper. Reference 14 is a doctoral thesis. The correction is highlighted in blue. We appreciate for your warm work earnestly, and hope that the correction will meet with approval.

Round 2

Reviewer 1 Report (Previous Reviewer 2)

Dear authors, you substantially considered my previous remarks. So I guess that no further improvement is necessary.

This manuscript is a resubmission of an earlier submission. The following is a list of the peer review reports and author responses from that submission.

Round 1

Reviewer 1 Report

The paper is impressive. It proposes a novel camera calibration method combining physics and mathematics for low cost cameras in both air and underwater environments.

The paper is well writen, with minor errors listed as follows.

"to both in-air and underwater Environment." -> "to both in-air and underwater environments."

"in the air and underwater environment." -> "in the air and underwater environments."

"For example, engineering applications like the three dimensional image reconstruction for the important culture/religion sites [1], the deformation monitoring of the industrial components [2], topographic mapping of the shallow water area [3], and accurate underwater measurements of fish [4] and so on." -> this sentence is incomplete

"are introduced in to" -> "are introduced to"

"to both in-air and underwater Environment." -> "to both in-air and underwater environments."

"At the end, several" -> "In the end, several"

"parameters, act as the" -> "parameters, acting as the"

"using the Equation (2)" -> "using Equation (2)"

"owing to the isotropy of circular," -> "owing to the isotropy of circles,"

"and the two neighboring" -> "and two neighboring"

"a 3D test filed" -> "a 3D test field"

"established 3D test filed" -> "established 3D test field"

"The established 3D test filed" -> "The established 3D test field"

"distributed in the test filed" -> "distributed in the test field"

"shown in Figure 5-10." -> "shown in Figures 5-10."

"(Figure 5-7, Table 1)," -> "(Figures 5-7, Table 1),"

"(Figure 5-7)." -> "(Figures 5-7)."

"Figure 8-10, Table 1)," -> "Figures 8-10, Table 1),"

"(Figure 8-10)," -> "(Figures 8-10),"

"Figure 11-13" -> "Figures 11-13"

"(Figure 11-13, Table 1)," -> "(Figures 11-13, Table 1),"

"the light rays refracts" -> "the light rays refract"

"distortions are decreased to 0" -> "distortions are decreased to close to 0"

It would be nice to make a toolbox available so that other researchers could easly use the proposed calibration method.

Reviewer 2 Report

Your novel camera calibration strategy based on iteratively performed calibration process is interesting and may lead to considerable improvements of recent calibrations. However, it has one fundamental lack: it was not evaluated by independent measurement data. Hence, the presented achieved accuracy is uncertain. It would have been confirmed by independent measurement data.

Additionally, the application to underwater camera calibration is very obscure. According to the theory of refraction at different media, your calibration method can provide valid results only for one certain distance in the object room, i. e., for only one plane in the water. This is not usable for practical applications such as 3D measurements.

I suggest to delete the underwater part of the manuscript and complete the other part by independent measurement results and application examples.